# Analysis of the Spatiotemporal Changes in Selected Ecosystem Services Caused by Free Trade Initiatives in Incheon, Korea

**DOI:** 10.3390/ijerph20053812

**Published:** 2023-02-21

**Authors:** Jiyoung Choi, Sangdon Lee

**Affiliations:** 1Research Institute of Agriculture and Life Sciences, Seoul National University, Seoul 08826, Republic of Korea; 2Department of Environmental Science and Engineering, Ewha Womans University, Seoul 03760, Republic of Korea

**Keywords:** biodiversity and ecosystem services (BES), IFEZ (Incheon Free Economics Zone), urban development, land-use change, carbon fixation change

## Abstract

Expansion of a development area can negatively impact ecosystems by decreasing or fragmenting habitats. With increased awareness of the importance of biodiversity and ecosystem services (BES), ecosystem service evaluations have attracted increased attention. The geography surrounding the city of Incheon is ecologically valuable because of its ecological diversity, attributed to its mudflats and coastal terrain. This study analyzed changes caused by the Incheon Free Economic Zone (IFEZ) agreement in the ecosystem services of this area using uses the Integrated Valuation of Ecosystem Services and Tradeoffs model and assesses the impacts of BES before and after the implementation of the agreement. Results revealed that carbon fixation and habitat quality decreased by approximately 40% and 37%, respectively, owing to the development caused by the agreement (*p* < 0.01). Additionally, endangered species and migratory birds were not protected by the terms of the IFEZ, and a decline in habitats, prey, and breeding sites was observed. This study suggests that the value of ecosystem services and the expansion of conservation areas should be considered part of ecological research under economic free trade agreements.

## 1. Introduction

Urban expansion aimed at economic development is undermining biodiversity and ecosystem services (BES) [1,2]. Major detrimental impacts resulting from land-use changes include the reduction and fragmentation of wildlife habitat and the threat to numerous species and ecosystems [3,4]. The economic loss caused by these impacts on the Earth’s overall ecosystem services was valued at USD 33 trillion annually, and climate control and biodiversity-related ecosystem services continuously decline [5]. As unrestrained development damages natural environments and reduces biodiversity, the importance of natural capital management, which evaluates the ecological, social, and economic value of BES, has become increasingly evident [6,7,8,9].

Since 2005, ecosystem service studies focusing on biodiversity have been evaluated. Recently, climate change, natural assets, and water yield have also been evaluated for natural asset protection, ecosystem service management, and sustainability [6,10,11]. Ecosystem services assessment or ecosystem quantification has been performed using various models [11,12]. For example, tools such as Artificial Intelligence for Ecosystem Services, Integrated Valuation of Ecosystem Services and Tradeoffs (InVEST), Multiscale Integrated Earth Systems, and Ocean Health Index have been employed for the environmental assessment of terrestrial and marine ecosystems [12,13,14]. Recently, assessments have been performed using an integrated ecosystem services model that considers ecosystem functions with spatiotemporal diversity and complexity. Integrated ecosystem services assessment employs ecosystem indexes that consider the developmental stage of a country or region [15,16]. In addition, as land use changes caused by development cause various decreases in ecosystem services, studies have investigated using measurement techniques to ensure No Net Loss (NNL) of ecosystems in future environmental impact assessments [17,18].

Numerous studies related to ecosystem service have demonstrated that land use changes affect carbon storage, thereby reducing ecosystem services by affecting climate control via a reduction in water content and habitat quality owing to the degradation of animal habitat. By providing the value of changes in the value of ecosystem services [19,20,21].

This study analyzes changes in BES depending on land-use changes caused by urban development. Korea’s Free Economic Trade Agreement was implemented in 2003 to attract foreign investment, strengthen national competitiveness, and improve productivity by making the country a business hub for northeast Asia. Incheon was the first area designated a Free Economic Zone in 2003. Its development plan for investment maintenance ran from 2010 to 2014, and a period of development completion ran from 2015 to 2020 (Table 1).

Several studies have reported that the subject areas of this study require ecological research in relation to the free economic trade agreement [22,23,24]. Although there are considerable social and economic benefits resulting from the Incheon Free Economic Zone (IFEZ) agreement, the ecosystem of the study area has not been protected. In addition, the ongoing development of mudflats and ocean reclamation has resulted in a decrease in available food for migratory birds, small fish, lugworms, and crustaceans [25,26]. The loss of feeding grounds has forced the relocation of birds, resulting in the unprotection of internationally endangered birds. These conditions are yet to be officially recognized, and the severe damage caused by sustained urban development is increasing [26,27,28]. The study area investigated in this study included migratory bird arrivals and wetland protection areas, and the high ecological value of this area has been reported in previous studies. It is an area where the expected reclamation of tidal flats and urban development plans until 2025 are planned, and it is an area that requires a scientific evaluation of the effect of the agreement on the ecosystem. Therefore, the present study was undertaken to analyze changes in ecosystem services caused by the IFEZ.

## 2. Material and Methods

### 2.1. Study Area

The IFEZ, which was introduced on 11 August 2003, covers three zones according to the project purpose of each district: (1) Songdo International City, which was to host international business development; (2) Yeongjong International City at Incheon International Airport, which was to serve as a core base for aviation logistics; and (3) Cheongna International City, which was designated as a location for international finance. Because it is in the geographical center of northeast Asia, Incheon is an optimal location for an international hub city, comparable to Hong Kong and Singapore. This agreement has a 3-step development plan: (1) Area extension from the reclamation; (2) Acceleration of the development by investment attraction; (3) Completion of the development [29].

The study area, however, has a unique topography of mudflats, wetlands, and islands. The study area’s western border is characterized by ecologically diverse topography featuring wetlands, mudflats, and islands. A “Detailed investigative study on biodiversity” conducted nine times by the National Institute of Biological Resources of South Korea’s Ministry of Environment found that there are 161 species, including mollusks, bryozoans, bryophytes, insects, birds, and mammals in the region [30]. In addition, it is home to 267 (17.2%) of Korea’s endangered species, indicating that it is a biologically important area for endangered wildlife. Furthermore, the study area (i.e., Incheon Metropolitan City) has the second-highest discovery rate of endangered species in South Korea [22,23]. The discovery rate of endangered bird species accounts for 70.4 percent among all endangered species, which is higher than the discovery rate of any other type of endangered species (mammals, birds, amphibian reptiles, insects, invertebrates, plants, algae, and seaweed (Table 2)). It incorporates major habitats for migratory birds along the East Asian-Australian Flyway.

The remaining areas are important under the Ramsar Wetlands Convention and habitats for migratory birds. It is also a group breeding site for domestically listed endangered species, included in the International Union for Conservation of Nature (IUCN) Red List, such as Saunders’ gulls (*Chroicocephalus saundersi*), oystercatchers (*Haematopus ostralegus*) and little tern (*Sterna albifrons*). Therefore, it has been legally mandated to investigate the impact of IFEZ on the habitats of endangered species and migratory birds.

After the IFEZ designation in 2003, urbanization progressed rapidly through large-scale reclamation projects. The study area is a biologically important region in which a range of research has been conducted, including studies on zoobenthos around Yeongjongdo Island [22], an ecologically important wetland protection area and a waterfowl habitat [27], and the distribution of the zoobenthos that provide food for waterfowl arriving at the mudflats and feeding grounds [28]. In addition, the area includes open ocean and islands on which amphibians and reptiles can be found. It is partly neighbored by ecologically rich military installations with well-conserved natural ecosystems. 

Therefore, the need to protect coastal wetlands has increased owing to the impact of large-scale reclamation efforts allowed in the IFEZ [23,27,28]. Thus, it is necessary to quantify ecosystem services associated with the urban development allowed by mudflat and seashore reclamation after the IFEZ designation. Therefore, the spatial study range included the Yeonsu-gu, Su-gu, and Jung-gu subunits within Incheon, areas affected by development after receiving the IFEZ designation (Figure 1). Environmental services were examined yearly to compare the pre- and post-IFEZ periods. In addition, statistical differences between the pre-IFEZ (1980–1990) and the post-IFEZ (2000–2018) periods.

### 2.2. Study Methods

InVEST model is a part of The Natural Capital Project (NatCap) and was used to identify correlations between natural capital and economic value in the study area. The revealing ecosystem services assessment results obtained using InVEST can support important policy decisions [31]. The advantages of this model include its flexible structure in time and space, which enables timely scenario analyses, as well as global, national, and regional analyses. In addition, this model facilitates the acquisition of input data based on land use and considers natural and environmental values in the decision-making process between development and conservation [31,32]. In this study, the model was designed to quantitatively examine the changes in the ecosystem using two indicators: (1) estimated changes in biodiversity resulting from observed land use modifications and (2) changes in carbon storage.

Urban development causes land use changes, which affect biodiversity and ecosystem services on a diverse scale in time and space. In this study, the model was designed to quantitatively examine the changes in the ecosystem through the ecological service measured by the carbon analysis and biodiversity evaluation based on the change in land use. In the MA report [33], ecosystem services are considered important enough to be defined as the basis for biodiversity and its resiliency, and ecological service indicators are considered important factors in national and regional development policymaking [34].

Recent studies on the InVEST model have employed overlapping evaluations by applying multiple appropriate ecosystem services in the research area [19,34,35,36,37]. This study applied the InVEST model to analyze the impact of IFEZ designation associated with a decrease in BES. It evaluates changes in the volume of ecosystem services using carbon storage as a climate change adaptation index and habitat quality revealed by land cover data as a biodiversity index [38]. This study compared spatiotemporal changes in diversity and carbon storage before and after the implementation of the IFEZ. Among the InVEST models, the Carbon model was used in this study to measure changes in carbon storage (a climate change adaptation index) resulting from land-use changes. The InVEST Habitat Quality (a biodiversity index) model was applied for overlapping analysis. As the study area includes mudflats and islands that play significant ecological roles for migratory birds and a number of endangered species, data on migratory bird habitats and the frequency of discovery of endangered species were added as spatial factors and analyzed in an overlapping manner.

#### 2.2.1. InVEST Carbon Model

The InVEST Carbon model can evaluate the amount of carbon fixed by an area. Its economic value can be estimated using carbon pool data and a land-use cover map [36]. The InVEST Carbon model estimates future and current values and analyzes changes and trends in ecosystem services based on a variety of spatial and temporal data. Carbon storage on a land area largely depends on the sizes of four carbon pools: above-ground biomass, below-ground biomass, soil, and dead organic matter. The InVEST Carbon model aggregates the amount of carbon stored in these pools according to land use maps and classifications [36,39] and the Carbon. Storage is calculated as the sum of the four carbon pools and can be expressed as follows (Equations (1) and (2)):(1)$$C_{i}=C_{i,A}+C_{i,B}+C_{i,S}+C_{i,D}\;$$
(2)$$C_{\mathrm{total}}=\sum C_{i}\times A_{i}$$
where i: land use type; $C_{i}$: carbon storage per unit area of land use type; $C_{i,A}$: above-ground carbon; $C_{i,B}$: underground carbon; $C_{i,S}$: soil carbon; $C_{i,D}$: dead organic matter; $C_{\mathrm{total}}$: Total carbon storage; $A_{i}$: Area based on land use type.

These equations can be applied to analyze ecosystem services according to changes in land use and measure ecosystem services for protected species [4,25,26]. The input data for this model, including land use maps drawn in 1980, 1990, 2000, and 2018, and a carbon pool table consisting of the above-ground mass, below-ground mass, soil, and dead mass, were prepared. The data were assembled considering factors presented in NatCap and preceding papers [19,36]. This study utilized data reported in previous studies in Korea for biomass input data [38,40,41]. To enhance the accuracy of the input data, this study applied the value of the forest growing stock per unit area by forest type multiplied by the carbon storage factor for the above- and below-ground biomass values [39]. The amount of carbon fixed in the study area, estimated using the input data and factors, was expressed as Mg C (Figure 2, Table 3). The InVEST Carbon model operates under one condition: the addition and loss of the carbon stock in the carbon cycle by land use are both zero. As carbon is fixed through the interaction of carbon sinks, such as the ocean, atmosphere, and land, it may affect the results of the model. Thus, this model assumed that the amount of carbon in each carbon sink is fixed and that the effect would not be as significant as that of the amount fixed in each carbon cycle.

In addition, an environmental service analysis using this model can be conducted on the economic value or social costs according to the change in the carbon fixation once additional data on the carbon discount rate and carbon price in each country are input after the carbon storage is analyzed. The economic value can be equated using the social costs that can be avoided by not releasing carbon into the atmosphere [19,36]. 

Land-use maps, common inputs for the model, employed geospatial data on the 1:25,000 mid-level classifications created by the Environmental Geographic Information Service (EGIS). The input data were converted to raster files required by the InVEST model using the ArcGIS program version 10.5 (Esri, Redlands, CA, USA).

#### 2.2.2. InVEST Habitat Quality Model

Biodiversity is intimately linked to the production of ecosystem services. Biodiversity patterns can be estimated by analyzing maps of land use and land use map in conjunction with threats to habitat. The habitat quality model for biodiversity aims to estimate the extent of habitat and vegetation types across a landscape and their state of degradation [19,31,36,40,42]. The Habitat Quality model was designed to evaluate biodiversity. A habitat quality index assigns values between 0.0 and 1.0, with higher values indicating greater biodiversity. This mechanism can be explained using the following formulas and four types of input data [40,41,42,43]. The first data point is the relative threat of a selected source; the second is the maximum impact distance between habitats and sources of the threats (the impact of a source decreases with an increase in the distance). The reduction appears as linear patterns (3) and index patterns (4). The linear exponential was applied when the influence of the threat factor consistently decreased as the distance from the habitat increased. In contrast, the exponential index can be applied when the influence of the threat factor decreases dramatically as the distance from the habitat increases. This study used the exponential index to analyze the urban area and the bare land, which drastically affected the influence of land use, and applied the linear exponential to the road areas which undergo linear development. 

The variable r indicates the threat degree between grid cell x and grid cell y, and d_xy_ indicates the maximum impact distance between a habitat and each threat source. d_r max_ indicates the maximum impact distance between a habitat and a threat source r.
(3)$$i_{\mathrm{rxy}}=1-\left(\frac{d_{\mathrm{xy}}}{d_{r\;\max}}\right)$$
(4)$$i_{\mathrm{rxy}}\;=\;\exp\left(-\left(\frac{2.99}{d_{r\;\max}}\right)d_{\mathrm{xy}}\right)\;$$

The third input data point is the habitat suitability index (HSI) for each factor, and the final data point is the degree of sensitivity of each habitat type to each threat source. Habitat quality can be calculated using values of the extent of habitat destruction. The potential for mitigating the threat and describing the level of protection can be calculated using Equation (5).
(5)$$D_{\mathrm{xj}}=\overset{R}{\underset{r=1}{\sum }}\overset{Y_{r}}{\underset{y=1}{\sum }}(\frac{w_{r}}{\sum_{r=1}^{R}w_{r}})r_{y}i_{\mathrm{rxy}}\beta_{x}S_{\mathrm{jr}}$$

The four kinds of input data converge to deliver an HSI in (6). Q_xj_ indicates a valuation of habitat quality to land-use map, type j.
(6)$$Q_{\mathrm{xj}}=H_{j}\left(1-\left(\frac{D_{\mathrm{xj}}{}^{z}}{D_{\mathrm{xj}}{}^{z}+k^{z}}\right)\right)$$

Data from NatCap [36] and previous research [40,41,42,43,44,45] from South Korea were used as reference material to select input data for this study. The expansion of urban and barren areas was selected as a threat source, along with road construction associated with forest fragmentation. Data on urban districts and barren lands were extracted from land-use maps [46]. Data on roads were extracted from road network documentation provided by the Ministry of Land, Infrastructure and Transport [47]. Among the mid-level classifications, we classified codes 110, 120, 130, 140, and 160 as urban areas; 150 as road areas; 210, 220, 230, 240, and 250 as farmland; and 610 to 620 as barren lands. The half-saturation constant, an additional input coefficient, was 0.5, as presented in previous studies [40,42,43,45]. The model provides numerical values for biodiversity that are qualitatively derived without units, and qualitative comparisons of changes in the amount of biodiversity may be conducted. ArcGIS Map 10.5 was used for all area analyses, and each factor was established in the Raster form under each model type (Table 4).

### 2.3. Analysis of Spatio-Temporal Correlation in Ecosystem Services Results

We analyzed the spatio-temporal correlation in the ecosystem services result. To this end, a statistical method was employed to determine whether the change in the ecosystem service result value was significant. The trend of the results over time and the trend with the Ministry of Environment’s data on habitat quality were determined using a statistical program (IBM SPSS Statistics 25). First, the Mann-Whitney U-test of the non-parametric statistical test was applied to identify statistically significant differences in the results before and after the IFEZ agreement. In addition, a non-parametric statistical test was conducted to judge the normality of the data. This data is not normally distributed and exhibited a non-linear pattern; thus non-parametric statistical test was applied.

An Ecological Natural Map (ENM) created by National Natural Environment Survey (NNES) was employed to identify correlations with habitat quality [40,42,45]. Because ENMs are based on biological survey data that enable the identification of an integrated ecological grade for a region, correlation frequency analysis was performed using the results of habitat quality and the bird-discovery points provided by the ENM. As a graded map based on diverse environmental information, an ENM does not provide precisely the same results as the InVEST model but shows a similar tendency. Therefore, we expected that bird discovery spots would be associated with high-quality habitats and confirmed that there was a correlation between the results of the NNES by the Ministry of Environment and the results of our study. The NNES is a nationwide survey conducted on a five-year basis pursuant to Article 30 of the Natural Environment Conservation Act of the Ministry of Environment [48].

The first NNES was performed in 1986 on topography, vegetation, plant, benthic macroinvertebrates, amphibians, reptiles, fishes, terrestrial insects, birds, and mammals [42,48]. In addition, ENM was produced pursuant to Article 34 of the *Natural Environment Conservation Act*. According to the Ministry of Environment in South Korea, the ENM classifies natural environments, including mountains, rivers, streams, lakes, farmland, and urban and inland wetlands, by grade according to their ecological value, natural quality, and landscape value. The ENM includes Grade 1, Grade 2, and Grade 3 classifications (Table 5). Grades 1 and 2 are natural environments that must be preserved, and where only minimal utilization and development are permitted. The ENM was based on biological survey data, which enabled us to identify the integrated ecological grade of a region and determine whether a significant correlation existed between the points of the discovery of birds and habitat quality.

## 3. Results and Discussion

### 3.1. Impacts of Changes in Land Use following IFEZ Introduction

The area affected and the rate of land-use change annually are presented in Figure 3. In 1980, wetlands accounted for the largest portion, followed by agricultural and forest areas. Owing to the construction of Incheon International Airport, Yeongjongdo Island had the greatest wetland area (31.3%) in coastal areas before reclamation. Between 1980 and 1990, the area under urban development experienced a major change, doubling, whereas the area covered by wetlands decreased by 21%. In 2000, the area of barren lands increased significantly to 82.62% compared to that of the previous decade. In 2018, urban and barren-land areas accounted for 46.26% of the total area, confirming the increase in urbanization over time.

### 3.2. Carbon Fixation Valuation

The amount of carbon stored in the study area tended to decrease over time. To measure the change between the pre-IFEZ (1980–1990) and post-IFEZ (2000–2018) periods, the changes were based on the amount and rate in a time series (Table 6). The amount of stored carbon in 1990 decreased by approximately 10% compared to that stored in 1980, corresponding to a 201,328 Mg C decrease. Carbon storage in 2018 decreased by twice the rate in the pre-IFEZ era, resulting in a loss of 1,176,650 Mg C, a decline of approximately 28% compared to 2000. This amounted to an overall annual decrease of 1.49%, which was 0.56% greater than before the designation of the IFEZ, reflecting a rapid reduction in carbon storage.

Maps of the distribution and changes in carbon storage by region are presented in Figure 4. The empty space in the 1980 map is Section 11 of Songdo International City before it was reclaimed. In the 1980 map, most areas are dark blue, indicating high carbon storage. Prior to development, a considerable amount of carbon was stored in the areas that were later developed into Songdo, Yeongjong, and Cheongna International cities. Jung-gu in the western part of Incheon exhibited a significant increase in the land area owing to reclamation for the phased expansion of Incheon International Airport in 2008. Still, the carbon storage of most of this area was low, ranging from 0.0 to 19.29 Mg C. Seo-gu, in the northern portion of Incheon, was the only portion of Cheongna International City that exhibited a decrease in carbon storage over time after development. The former downtown area of Yeonsu-gu, in the southern section of Incheon, was developed into a manufacturing and residential area. After being designated as Songdo International City, its land area increased by 53.45 km^2^ through the large-scale reclamation of the sea and mudflats. An open ocean area within Songdo is slated for future reclamation. 

### 3.3. Habitat Quality

Urban and agricultural areas were treated as threatening facial factors, and roads as lineal. Roads were classified as large, medium, or small. Annual road boundary data were extracted from seamless digital topographic maps from the National Geographic Information Institute. The input data was converted into an appropriate format using the Habitat Quality model, which shows the habitat quality regarding biodiversity. The figures are unitless but were accompanied by quantitative changes.

According to the model, the habitat quality from 1980–1990 (pre-IFEZ) decreased by 0.72%, resulting in an annual biodiversity decrease of 0.07%. Habitat quality in 2000–2018 (post-IFEZ) decreased by 15.09%, corresponding to an annual decline of 0.79% (Table 7).

Figure 5 maps the periodic results of the habitat quality. The changes in the amount of habitat quality by region were confirmed in this study. Most of the areas on the map in 1980 were dark red, indicating high habitat quality. In addition, the habitat quality of the areas set to become Songdo, Yeongjong, and Cheongna International cities was high. Particularly, the color difference in the habitat quality maps from 1980 and 1990 compared to those of 2000 and 2018 was significant.

### 3.4. Statistical Analysis of Changes in the Ecosystem Services over Time

We analyzed whether the ecosystem services changes caused by the IFEZ designation were significant. To this end, the trends from 1980 to 2018 were confirmed using statistics (processed in IBM SPSS statistics 25). The changes between the pre- and post-IFEZ periods were statistically verified to identify the effects of urban expansion caused by the designation. The R^2^ values for carbon storage and habitat quality between 1980 and 2018 were 0.923 and 0.889, respectively. A non-parametric Mann–Whitney U test was performed for the data obtained before and after the designation, and the results revealed that the ecosystem services changed significantly (*p* < 0.01).

### 3.5. Correlation between Habitat Quality and Ecological Natural Maps

The average quality of habitats was calculated based on data from the discovery points of birds [36] using the fourth (and most recent) NNES (2014–2018) overlapped with the habitat quality results for 2018. The comparison of the habitat quality of the bird discovery points with the discovery points of endangered birds (Figure 6) revealed that the average quality of bird habitats and those of endangered birds were 0.71 ± 0.19 and 0.79 ± 0.22, respectively, indicating that the habitat quality of most discovery points was excellent (Figure 7). The discovery points of the three major endangered species (Saunders’ gulls, oystercatchers, and black-faced spoonbills) in the study area are presented in Figure 6, along with the habitat quality distribution criteria. A Pearson chi-square test revealed that there was a statistically significant (*p* < 0.05) difference in the habitat quality of the three endangered species, and among the endangered species, black-faced spoonbills were spotted in places with high-quality habitats.

In addition, the Pearson chi-square test confirmed that the correlation between habitat quality and ENM was statistically significant (*p* < 0.01) (Table 8). Grade 1 ENMs are places of high natural value, which were distributed close to 1.0, indicating a high-quality habitat. Grade 3 was distributed close to 0.0, indicating a low-quality habitat (Figure 8). None of the study areas were distributed in an ENM separate area, and the ENM grade negatively correlated with the habitat quality. 

## 4. Conclusions

In this study, ecological changes induced by the IFEZ designation were evaluated based on an ecosystem services valuation using two InVEST models: (1) quantitative evaluation of carbon storage according to land-use change and (2) qualitative evaluation of biodiversity through habitat quality analysis. 

The results revealed that carbon fixation was reduced by 40% owing to the urban development caused by the agreement, which resulted in a 793,586.25 Mg of C decrease in carbon storage. In addition, the habitat quality values e reduced by 0.2, corresponding to a decrease of approximately 37%. Further, the statistical analyses conducted before and after the agreement confirmed the statistical significance of the decrease in the ecosystem services caused by the agreement. 

The damage to bird habitats owing to the reclamation and construction of the tideland and seashores of the development area has raised concerns regarding the ecosystem. Thus, this study employed the InVEST model to obtain quantitative results of the ecological repercussions predicted in previous studies. 

Nevertheless, there were limitations to the assumption for operating the model, along with other factors that should be considered in future research. First, the input figures on the carbon pool table might change depending on the referenced existing study. To obtain more accurate carbon fixation values, a more detailed and categorized calculation of carbon fixation value per unit area should first be conducted. Second, the carbon model displayed limitations regarding the factors affecting the amount of carbon fixation in the land. This model represented the carbon fixture concerning the carbon pool table, input data, and changes in land use, which caused difficulties in identifying temporary carbon fixation changes in the soil. Therefore, when using this model to estimate carbon fixation, the macroscopic changes in land use and microscopic changes, such as autogenic succession, should be adjusted by conducting a temporal segmentation analysis on the input data.

Future research on the model can also secure individual input data according to the target species in the research area by applying the habitat suitability index as the input data of the Habitat Quality model. The accuracy of this model can be further improved if the limitations of this research are addressed. 

Although quantitative and predictive evaluation is difficult owing to the nature of the ecosystem, this model is one of the scientific techniques for evaluating the ecosystem in situations that continuously require such methods [5,6,7,49,50]. Related research is essential for the preservation and continued use of the ecosystem and ecosystem services. 

Lastly, ecological changes associated with IFEZ designation should be examined, and maintenance and management of carbon storage to restore climate controls and the valuation of biodiversity should be considered by developers and government officials. In natural environment ecology, the EIA for development, biodiversity, dominance, natural environment, cultural assets, and protected species should be investigated. Applying the quantitative evaluation methodology of ecosystem services for the planned development area is expected to be meaningful.

Furthermore, the ecosystem service methodology can be applied to evaluate ecosystem changes affected by external influences, such as development projects. It is believed that this technique can be a methodology for the sustainable development of harmonious economic, social, and environmental development through the evaluation and prediction of future ecosystem services using scenarios.

In addition, it can be used as basic data in making development policy decisions through qualitative and quantitative evaluation along with other environmental fields by utilizing AI and ICT methods, which are related methodologies according to the 4th industrial revolution.

## Figures and Tables

**Figure 1 ijerph-20-03812-f001:**
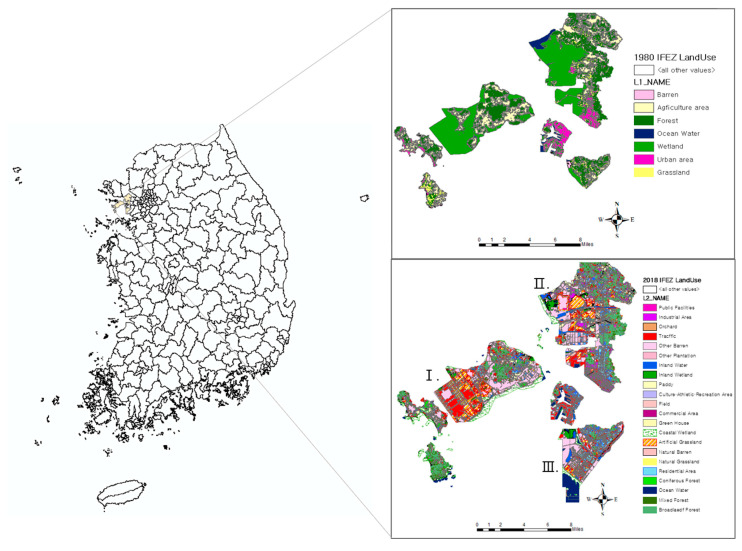
Study area. Ⅰ: Yeoungjong International City, Ⅱ: Cheongna International City, Ⅲ: Songdo International City.

**Figure 2 ijerph-20-03812-f002:**
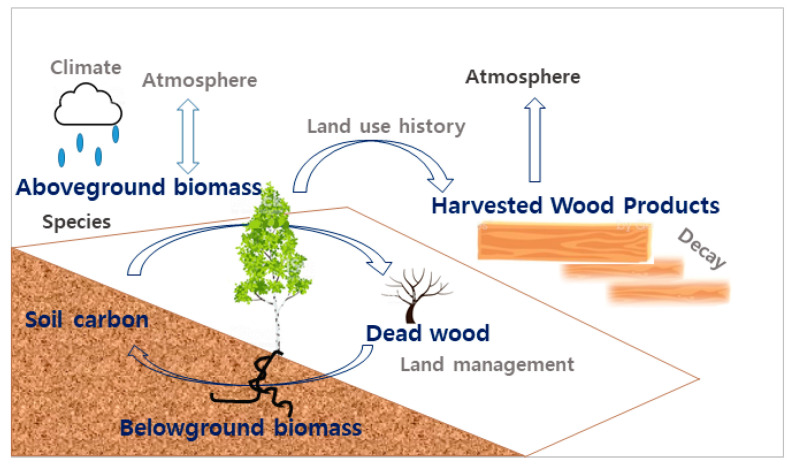
InVEST Carbon model process.

**Figure 3 ijerph-20-03812-f003:**
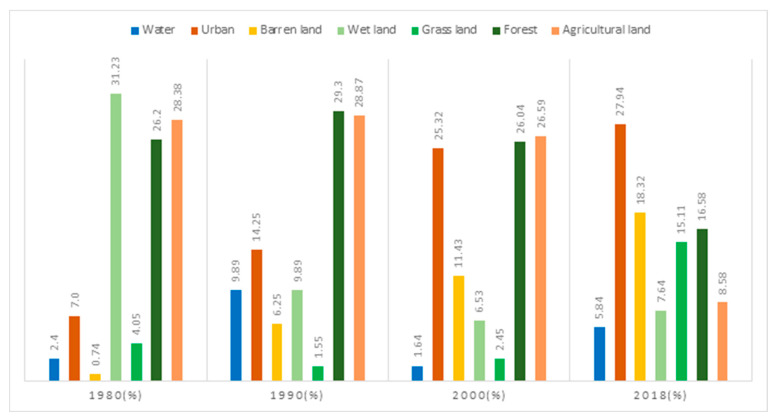
Total land-use class rate from 1980–2018.

**Figure 4 ijerph-20-03812-f004:**
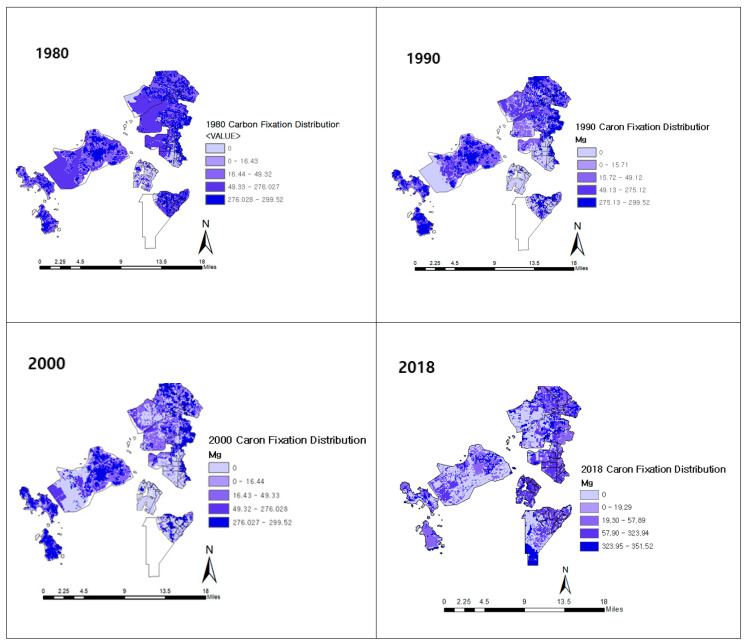
Carbon fixation from 1980 to 2018. The white areas (in the bottom right image) were not reclaimed until 2018.

**Figure 5 ijerph-20-03812-f005:**
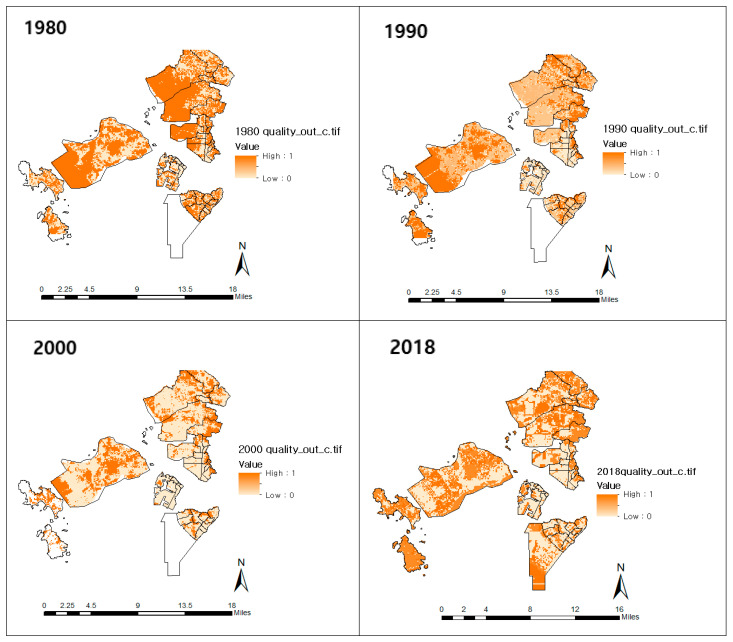
Habitat quality distribution from 1980 to 2018. The white areas (in the bottom right image) were not reclaimed until 2018.

**Figure 6 ijerph-20-03812-f006:**
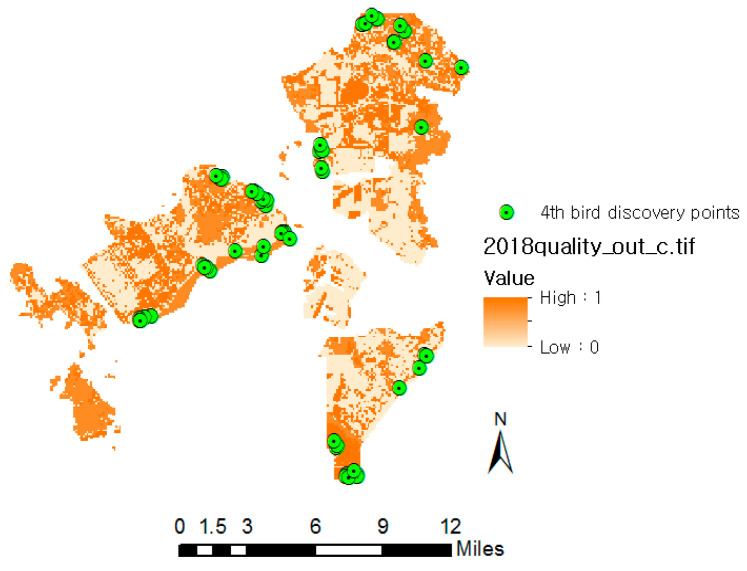
Overlapping maps of bird spotting points and 2018 habitat quality.

**Figure 7 ijerph-20-03812-f007:**
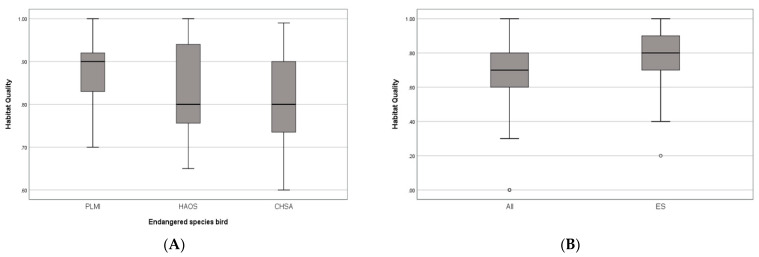
Habitat quality. (**A**): All bird points (n = 261), (**B**): Endangered bird species. PLMI: *Platalea minor* (n = 12); HAOS: *Haematopus ostralegus* (n = 35); CHSA: *Chroicocephalus saundersi* (n = 27). *p* < 0.05. Box: 25–75%; line: median; whiskers; minimum and maximum values; circles: individual outliers.

**Figure 8 ijerph-20-03812-f008:**
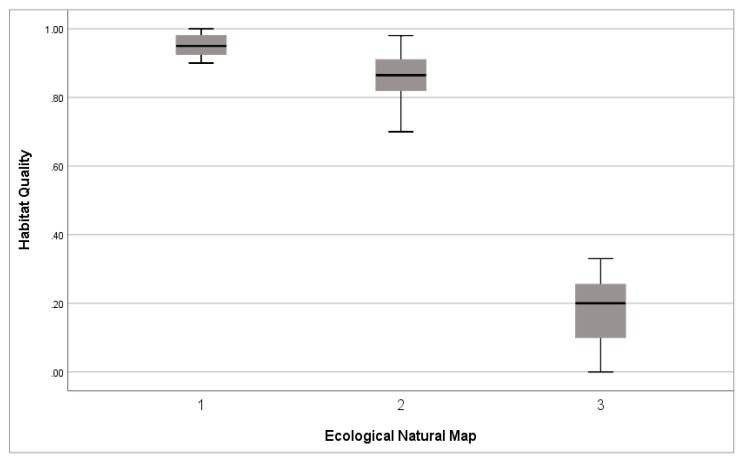
Correlation between the ecological natural maps and habitat quality (*p* < 0.01). (Box: 25–75%; line: median; whiskers: minimum and maximum values).

**Table 1 ijerph-20-03812-t001:** Status of Korean Free Economic Trade Zones.

Period (Launch)	Zone Name	Location	Area (km^2^)	Development Period
Ⅰ (2003)	Incheon Free Economic Zone (IFEZ)	Incheon (Yeonsu, Jung, Seo-gu)	123.49	2003–2020
	Busan-Jinhae Free Economic Zone (BJFEZ)	Busan (Gangseo-gu), Jinhae (Jinhae)	51.06	2003–2020
	Gwangyang Bay Area Free Economic Zone (GFEZ)	Jeonnam (Yeosu, Suncheon, Gwangyang) Gyeongnam (Hadong)	69.57	2004–2020
Ⅱ (2008)	Yellow Sea Free Economic Zone (YESFEZ)	Cheongnam (Dangjin, Asan, Seosan) Gyunggi (Pyungtaek, Hwasung)	4.36	2008–2020
	Daegu-Gyeongbuk Free Economic Zone (DGFEZ)	Daegu, Gyeongbuk (Gyeongsan, Yeongcheon, Pohang)	18.45	2008–2022
Ⅲ (2013)	Chungbuk Free Economic Zone (CBFEZ)	Chungbuk (Cheongju)	4.88	2013–2020
	East coast Free Economic Zone (EFEZ)	Gangwon (Donghae, Gangneung)	8.86	2013–2024

**Table 2 ijerph-20-03812-t002:** Comparison of the numbers of endangered species and endangered birds in Korea (Ministry of Environment 2018).

Region	Numbers of Endangered Bird Species	Numbers of Total Endangered Species	% (Endangered Bird Species/Total Endangered Species)
Seoul	5	14	35.71
Busan	25	41	60.98
Daegu	-	20	0.00
Incheon	162	230	70.43
Ulsan	-	14	0.00
Gyeonggi-do	184	450	40.89
Sejong	-	5	0.00
Gangwon-do	45	766	5.87
Chungbuk-do	9	175	5.14
Chungnam-do	89	269	33.09
Jeollabul-do	54	422	12.80
Jellanam-do	65	441	14.74
Gyeongbuk-do	7	266	2.63
Gyeongnam-do	33	323	10.22
Jeju	108	505	21.39
Gwangju	-	10	0.00
Daejeon	1	4	25.00
Total	787	3955	19.90

**Table 3 ijerph-20-03812-t003:** Input data and Format of InVEST Carbon model.

Input Data	Format	Source and References
Land-use map	Raster file (.tiff) 1:25,000 Land-use map	ArcGIS Map ver.10.5., [31,37]
Carbon Pool table	Table (.csv)	[35,38,40]

**Table 4 ijerph-20-03812-t004:** Input data and Format of the InVEST Habitat Quality model.

Input Data	Format	Source and References
Land cover map	Raster file (.tiff)	ArcGIS Map ver. 10.5., [31,46]
Threats files	Raster file (.tiff)	
Threats data	Table (.csv)	[31,36,41,42]
Sensitivity and habitat suitability Index data	Table (.csv)	
Half saturation constant	Number (0.5)	

**Table 5 ijerph-20-03812-t005:** Definitions of the ecological natural map grades [48].

Grade	Definition
1	Endangered species, main habitat for animals and plants; areas with particularly excellent ecosystems and landscapes
2	Area equivalent to Grade 1; areas worth protecting in the future; outside Grade 1
3	Areas subject to development or use

**Table 6 ijerph-20-03812-t006:** Change in Carbon Fixation from 1980–2018.

Carbon Fixation		Estimate per Year			
		1980	1990	2000	2018
Mg C		1,970,237.00	1,768,909.00	1,641,849.25	1,176,650.75
1980–1990		2000–2018		1980–2018	
Change	%	Change	%	Change	%
−201,328.00	−10.22	−465,198.50	−28.33	−793,586.25	−40.28

**Table 7 ijerph-20-03812-t007:** Change in Habitat Quality value from 1980–2018.

Habitat Quality		Estimate per Year			
		1980	1990	2000	2018
Biodiversity (unitless)		0.539 ± 0.22	0.535 ± 0.20	0.397 ± 0.19	0.337 ± 0.19
1980–1990		2000–2018		1980–2018	
Change	%	Change	%	Change	%
−0.0039	−0.72	−0.06	−15.09	−0.2	−37.48

**Table 8 ijerph-20-03812-t008:** Correlation coefficient between the ecological natural map and habitat quality.

		Ecological Natural Map	Habitat Quality
Ecological Natural Map	Pearson correlation	1	−0.845 **
	Statistical significance probability		0.000
Habitat Quality	Pearson correlation	−0.845 **	1
	Statistical significance probability	0.000	

** *p* < 0.01 (two-tailed).

## Data Availability

Not applicable.

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
