# Peer review of "Analysis of the Spatiotemporal Changes in Selected Ecosystem Services Caused by Free Trade Initiatives in Incheon, Korea"

_ijerph, 2023, doi:10.3390/ijerph20053812_

Round 1
Reviewer 1 Report (Previous Reviewer 1)
This research is dealing with spatiotemporal changes in ecosystem services in city of Incheon Korea as the results of free trade initiatives.
The study is very meaningful in its subject. The research method and analysis are also sound and well fit to the research subject.
Based on the above facts, I evaluate the study is currently proper for publication in present form.
Author Response
Thank you for your reviewing. We are most thankful for your comments.
Reviewer 2 Report (New Reviewer)
This study is interesting and especially valuable for ecosystem conservation. However, I found there are still many problems in the manuscript, which need to be further modified and improved.
1. The scientific problem of this manuscript is not clear, that is, the scientific problem need further to be summarized.
2. Some important results are not discussed in depth or not discussed. In this study,authors often talk about phenomena only according to the results, and the scientific problems behind the phenomena are not deeply discussed. Thus, I suggest the authors conduct an in-depth discussion according to the main results. Only in this way, the scientific value of the article can be well reflected. And which would bring certain inspiration for the future readers.
3. The uncertainty of InVEST model is great, for example, the selection of carbon pool is often empirical. Therefore, the limitations of the study should be further additionally issued.
4. The conclusions of the manuscript are too verbose and need to be condensed.
5. The sentence consistency of the manuscript is poor, for example, there is no logical relationship between the first and second sentences of the first paragraph in Introduction, and so on. Therefore, it is necessary to verify and modify the logic between sentences to make the article more fluent.
6. Many introductions about the Study area in the section of Introduction can be moved to the section of "2.1. Study area".
7. The execution process of the InVEST model does not need to be described in detail, but the simulation principle of the model should be further elaborated.
8. Lines 314-318, what is the relationship between habitat quality and carbon sequestration?
9. Lines 339-346,the statistical analysis methods should be introduced in the methods section, not here.
10. Check the full text for stupid mistakes, such as missing punctuation on line 146.
11. There are a lot of grammatical errors in sentences, so the manuscript needs polishing.
As a result of the above problems, I recommend the manuscript a major revision.
Author Response
Please find the attached file

Reviewer 3 Report (New Reviewer)
The title of the paper takes the signing of ‘free trade initiatives ’as the driving factor, but the introduction gives a large description of land use. How to quantify and characterize the relationship between free trade initiatives, urban development and land use? Is it reasonable for urban expansion to be represented only by land use change?
The introduction gives a large description of the research area, I suggest using concise sentences to express the importance and significance of the research area,and the rest parts can be specially described in the overview of the research area(2.1. Study area) in the second part(2.Material and methods).
Figure 6 shows the expression of habitat quality, but in line 315-317 expressed in terms of carbon storage. Is this reasonable?
Biodiversity and habitat quality are only part of ecosystem services, so, it is unreasonable to use the decline of biodiversity and habitat quality to represent the decline of the whole ecosystem services in line 407.
In my opinion, the natural ecological map is a form of data rather than a method, and the natural environmental survey is not organized by the author or his team. There are some tests (E.g., Pearson chi-square test, non-parametric Mann–Whitney U test) used in the results of this paper, but they are not described in the methods, which is unreasonable. It is suggested that the author should further restrict and express the relevant contents reasonably.
How can Table 7 show that ecological nature map grade is negatively correlated with habitat quality? Lines 381-382
The whole paper lacks in-depth discussion and analysis. In the section Results and discussion, the calculated values are simply described, and the actual situation of the study area, internal and external driving factors and the relationship between them are not discussed in depth. For example, in line 392, there is no in-depth correlation between biodiversity and habitat quality. At the same time, based on the research results, the paper does not put forward substantive policy recommendations for future ecosystem services, economic and social harmonious development in the study area. It is suggested that the author further optimize the full text.
Round 2
Reviewer 2 Report (New Reviewer)
I have reviewed the revised manuscript; I am satisfied that the issues raised in my original review have been addresses, and so I recommend acceptance.
Author Response
Thanks for the comments.
This manuscript is a resubmission of an earlier submission. The following is a list of the peer review reports and author responses from that submission.
Round 1
Reviewer 1 Report
The article is dealing with ecological and environmental impacts of IFEZ(Incheon Free Economic Zone) with proper methods and data. Since globally such economic agreements, like free economic zones, are widely spreading, to take proper evaluation on ecological and environmental impacts of such a economic development is very essentially required, and this study very well satisfies those needs. I hope, if possible, the authors can add expected usages of recent technological innovations, i.e. the 4th industrial revolution, for possible tools of ecological protection from economic development like IFEZ in the conclusion part with proper references, especially in Open Innovation (OI) perspectives. In general, I evaluate the study is very valuable, and the results of this study will give many meaningful policy insights to related areas and related players.
I recongize the strenth and weakness of the article as below.
(The strength of the article)
- It gives specific ecological impacts of a sepcific economic development (i.e. IFEZ) based on sound data and meaningful analysis. So, the article is very useful to the policymaker who has plan to build up such economic developments, but who also want to minimize ecological damages.
- The data and analysis is easy to understand and it is well explained using proper graphical illustration. Figures, those the authors are providing, are very easy to undersstand, so that they gives much more readability to the possible readers.
(The weakness of the article)
- It would be much better if the article can suggest possbile cures and ecological protections using current technological innovations (i.e. the 4th industrial revolution). The authors can use conclusion part to deal with such arguments.
In general, I recognize the values and meanings ot the article very good, and I evaluate the article is now proper for publication currently.
Author Response
REVIEWER 1
(The weakness of the article)
- It would be much better if the article can suggest possbile cures and ecological protections using current technological innovations (i.e. the 4th industrial revolution). The authors can use conclusion part to deal with such arguments.
In general, I recognize the values and meanings ot the article very good, and I evaluate the article is now proper for publication currently.
-L 320-324: we added the following statement :
In the natural environment ecology the EIA for development, biodiversity, dominance, natural environment, cultural assets, and protected species are investigated. It is judged that it will be a meaningful study by applying the quantitative evaluation methodology of ecosystem services for the planned development area.
Afterwards, it is judged that it can be used as basic data in making development policy decisions through qualitative and quantitative evaluation along with other environmental fields by utilizing AI and ICT methods, which are related methodologies according to the 4th industrial revolution.

Reviewer 2 Report
This paper tracks the environmental costs of rapid urbanisation in Incheon, Korea. The authors evaluate these costs in terms of carbon fixation and habitat quality indices that are extrapolated from land use and biodiversity data using the InVEST modelling suite.
Despite the clear interest and value in tracking the environmental, biodiversity and ecosystem service costs of rapid urbanisation, I do not feel that this study is ready for publication. In essence, the model appears to serve no purpose in deriving any of the conclusions, as the input data provides clearer values of the environmental losses caused by urbanisation (e.g. what is the added value of high uncertainty/no-sensitivity analysis extrapolation of carbon fixation when the input data identifies a fourfold decrease in forest cover and wetlands?). I feel that there is a lack of reflection as to the purpose of the model – it is applied as a black-box gimmick, with no clarity on what model assumptions are, how parameters are derived, what indices are for, what terms mean (see major comments below). Similarly, results are listed as a series of values, little apparent reflection to what they represent with regards to the initial questions or hypotheses, or what such values mean.
I think it is also important to reflect on the concepts used, such as ecosystem services. Here the ES framework is used as an excuse to convert land use data to carbon fixation terms – which grossly under-represents the multiple different services that are lost when forest cover is lost for example, or mudflats or wetlands… I’m sorry to say that this study probably contributes more to a problem of undervaluing the importance of nature than to flagging the problem.
Below are some more specific suggestions, but overall I recommend a rethink of the purpose and methods of the research.
Major comments:
Methods section only clear to InVEST modellers, needs to be developed to explain assumptions and processes modelled. E.g. in overlap analyses (what are they?) are you looking for correlations in different overlays and assuming causality? what are the assumptions used to guide a translation of carbon fixed to economic value? … How do these assumptions influence the interpretation of results? What does the model add to simple descrciptions of land use changes and biomass trends? What do the linear patterns and index patterns of the equations mean or represent? How is a habitat suitability index computed? Did the ministries who collected the data provide them in the format required for the equations or did you / Invest extrapolate them in some way. Again, what are the underlying assumptions?
It is not clear what value the InVEST modelling brings to the study, indeed, the input biodiversity data or land use maps seem to be sufficient to track the changes discussed in the conclusion (cc.f. lines 290-317).
In line 320, ‘ we were able to confirm a decline in ecosystems services stemming from habitat desctruction, loss of breeding sites and loss of wildlife prey…’ none of these conclusions come from the InVEST modelling , they do not reflect an economic valuation of ecosystem services (the usual InVEST purpose), and do not even stem from a discussion or analysis of ecosystem services. I think a map reading before and after, or following the trends in the data used as input to the model, would have provided as much information.
…
Minor comments:
- 14: favoring human development
- 68: what is meant by ‘s of which are’ – in grey letters?
- 71: measures of the impact of IFEZ on habitats….
- 76- 79: Unclear sentence / paragraph, which is unfortunate because it is an important sentence. I suggest breaking it down to several small sentences.
Figure 1: since the study tracks changes between 1980 and 2018, maybe a map of the land use in 1980 would be relevant to compare with the 2018 one here.
l.105-110: need a better intro and background to the InVEST model suite, their development, uses and assumptions.
- 110-115: need to explain what overlapping analyses are and what they are for.
- 138-144: what threat are we talking about?
l.171: second highest rate
Table 2:
the header row is truncated.
what time period does this table refer to?
Figure 3:
I do not understand what a clip frequency is
Are the study areas basically everything but the migratory birds’ habitat?
What do the pies represent: sum of what? What of birds?
- 187-188: what is the point of the discovery of birds?
Figure 4. the two end yellow boxes look truncated/ do not make sense to me: ‘evaluation of ecology and natural What? ‘ Making of Ecology and Natural what?’
- 195: largest portion of what?
Table 4: a graph might represent the table better.
Figure 5: a large dark blue area at the bottom of zone III (Songdo International City?) in the 2018 map indicates very high carbon storage, yet on figure 1 this area is described as ocean water. Why is the C fixation of ocean water that high in 2018 and not in previous time frames?
L 228: what are facial and lineal threatening factors?
- 237-238, why is the annual decline rate relevant?
Figure 6: am still mystified by the sudden high quality of the ocean tip of zone 3 in 2018, and apparent absence of data for that area before. If the area has been subjected to land reclamation, then this needs to be flagged in the map somehow, and it doesn’t explain to me the change in quality (carbon fixing and habitat) of the ocean water bit.
Author Response
Please find the attached file for reviewer #2